# Adaptation of the SBTool for Sustainability Assessment of High School Buildings in Portugal—SAHSB<sup>PT</sup>

**Tatiana Santos Saraiva** [1,*] **, Manuela de Almeida** [2] **and Luís Bragança** [2]

1   International Doctoral Program in Sustainable Built Environment, School of Engineering, Minho University, 4800058 Guimarães, Portugal
2   Department of Civil Engineering, School of Engineering, Minho University, 4800058 Guimarães, Portugal
*   Correspondence: saraivaus@yahoo.com; Tel.: +55-96-98106-1627

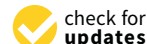

**Featured Application: The methodology proposed in this article, SAHSB<sup>PT</sup> methodology, should be applied specifically in Portuguese high school buildings, seeking to improve, verify and certify sustainability in these buildings, related to social, environmental and economic aspects.**

**Abstract:** The European Union objectives in the Horizon 2020 program aim to reduce environmental impact through strategies such as the improvement of energy efficiency and the use of renewable technologies. With regard to the goal of sustainable development—which integrates environmental, social, and economic dimensions relating to the preservation of the planet and the integrity of consumers—several types of sustainability certification tools are currently used in the construction industry e.g., Leadership in Energy and Environmental Design (LEED), the Building Research Establishment Environmental Assessment Method (BREEAM), and the Sustainable Building Tool (SBTool) There are international sustainability methodologies designed specifically for high schools and methodologies specific to the Portuguese reality, such as Natura Domus, LiderA (Liderar pelo Ambiente para a construção sustentável, Leading the Environment for Sustainable Construction) and SBtool<sup>PT</sup> (Sustainable Building Tool, Portugal). Currently, with the concern in Portugal to requalify schools, it has become necessary to develop a specific methodology for school buildings according to the Portuguese reality. This work highlights the SBTool methodology that is employed in several countries and can be adapted to basic education institutions as the basis for the formulation of responsible citizens and the development of a country. The main aim of this study is to adapt an already existing assessment tool of sustainability (SBTool<sup>PT</sup>), maintaining some indicators while modifying and adding others, in order to develop a methodology specifically for conducting a Sustainability Assessment of High School Buildings in Portugal—SAHSB<sup>PT</sup>. In order to achieve this goal, other methodologies that already incorporate parameters relating to the school environment are analyzed, such as LEED BD + C Schools (LEED Building Design and Construction School), BREEAM Education 2008, and the SBTool for K–12 schools.

**Keywords:** SBTool; school buildings; sustainability assessment tools; sustainability

---

## 1. Introduction

In the early 1970s, in light of mounting evidence and analysis on social development presented by the scientific community, it became increasingly clear that escalating rates of pollution in water, soil, and the atmosphere associated with the growth of the global population could have severe consequences for life on Earth. Thus, the concept of sustainable development was conceived and

was internationally recognized in 1972 in Stockholm at the first Earth Summit. Since then, general concern about the state of the environment and the potential consequences of the exponential increase in population—environmental pollution, resource consumption, and deforestation—has persisted [1].

Some advances have been made with the intention to increase sustainable construction, especially in the research and development of tools for the assessment of the sustainability of buildings. As a result of the use of these tools, it is possible to improve the sustainability of building construction [2]. These tools also distinguish between sustainable and unsustainable practices, thus facilitating conscientious decisions in projects and in construction phases.

It is possible to improve the sustainability of building construction while taking into account environmental, social and economic aspects. Regarding the environmental aspect, sustainability tools can reduce the life cycle environmental impacts and heat island effects of the building, mitigate the environmental impact from the destruction of forests, reduce energy and water consumption, and minimize the volume of construction and demolition waste. With regard to the social aspect, the methodologies can increase the environmental comfort, productivity and life quality of students; prioritize access to public transport and facilities; improve occupants' security and safety; raise awareness among children about the importance of sustainability and promote the use of transport systems. It can also reduce the construction and maintenance costs of school buildings.

Several countries have developed tools, such as the Sustainable Building Tool, (SBTool Canada), Leadership in Energy and Environmental Design (LEED, United States), Building Research Establishment Environmental Assessment Method (BREEAM, United Kingdom), Comprehensive Assessment System for Built Environment Efficiency (CASBEE, Japan), Haute Qualité Environnementale—High Environmental Quality (HQE, France), and the National Australian Built Environment Rating System (NABERS, Australia), that allow for sustainability assessments of buildings [3]. These tools were made with the purpose of being adapted to all types of constructions. However, the need for producing tools for specific buildings such as residences, offices, shopping centers, hospitals, and so on was gradually recognized.

In 2009, the Sustainable Building Tool, Portugal (SBTool$^{PT}$) was categorized into specific methodologies, such as SBTool$^{PT}$-H (Sustainable Building Tool, Portugal, Homes), which concerns residential buildings [4], and in 2014 was further categorized into methodologies for the Urban Plan, SBTool$^{PT}$—STPU [5]. The methodology for the Office Buildings [6] research project, SBTool$^{PT}$-STP Tools, SBTool Portugal for Office (Serviço) Buildings, Tourism (Turismo) Buildings and Urban Plan (Planejamento Urbano, [1], was presented at the International Conference SB13 (2013) in Portugal.

Other systems have been developed and adapted according to the Portuguese reality, needs and requirements, such as Natura Domus, LiderA, and SBtool$^{PT}$ [7]. None of these methodologies has any certification system specific to school buildings in Portugal.

BREEAM, SBTool and LEED methodologies have elaborated on specific systems for school buildings, such as BREEAM Education (2008), SBTool for K–12 schools and LEED BD + C: Schools (2013). This is important, because the environment of schools is unique. Students and teachers spend several hours a day surrounded by the school's environment, and therefore, this specific environment has a major influence on their quality of life.

Along with the maintenance of the high school buildings, the sustainable refurbishment of high school buildings, in terms of the quality of life of teachers and students, energy efficiency, reduction of emission of greenhouse gases, optimization of water consumption, better indoor air quality, improvement of health and learning conditions, and minimization of costs, is crucial [8].

The Program for the Modernization of Schools for Secondary Education (Programa de Modernização de Escolas de Educação Secundária, PMEES) aimed to place Portuguese education in a position of international reference. The PMEES included interventions in 332 of the 447 Portuguese high schools, equivalent to 74% of all high schools in Portugal. Empresa Parque Escolar (EPE), created in 2007 for this purpose, is carrying out the construction and renovation works of the high schools [9].

No sustainability assessment exists for school buildings that is adequate to the Portuguese reality and benefits the requalification of schools. This article aims to analyze the importance of the use of sustainability indicators to reach and develop strategies for the regeneration of school buildings, taking into account the sustainability assessment methodology of SBTool[PT], which is based on the international SBTool methodology developed by iiSBE (International Initiative for a Sustainable Built Environment). Adapting this methodology creates a rubric that is specifically applicable to the evaluation of the sustainability of high school buildings in Portugal, which is important, because there is currently no such specific tool elaborated in this country for this purpose. Thus, this work adapts the SBTool methodology for the Sustainability Assessment of High School Buildings in Portugal, creating a tool tailored for Portugal—the SAHSB[PT] methodology. SAHSB[PT] will assist architects and designers with the improvement of sustainability in high school buildings in the project design phase or the rehabilitation of existing buildings.

All building types have different characteristics; thus, it is necessary to develop specific sustainability methodologies to address those distinct traits and functions. In other words, a school building has certain particularities that must be considered with respect to implementing principles of sustainability; for instance, as it is an institution designed to provide learning spaces and learning environments for the teaching of students, children, and adolescents, it requires a more dedicated and rigorous level of comfort geared towards a younger population. Furthermore, it is important to plan construction in order to reduce environmental impacts by using the appropriate equipment and technology.

Several environmental assessment tools have been created in order to assist with the development of more sustainable building projects. Among these methodologies, there are some tailored to schools, such as LEED BD + C: Schools, BREEAM Education 2008, and the SBTool for K–12 schools, which promote greater sustainability in the built environment, such as in the project design phase or rehabilitation projects. The methodology proposed in this work is similar to SBTool[PT]-STP for Office Buildings and SBTool[PT]-H.

Some SBTool[PT]-STP for Office Buildings indicators have been modified, removed, or added here to better adapt it to school buildings in SAHSB[PT]. Indicators related to environmental comfort and security have been modified, because SAHSB[PT] must consider the needs of children and thus the values for reference practices must be stricter. Energy indicators have been modified to adapt to new legislation. Indicators concerning "sustainability awareness" and "ergonomic comfort" have been included to meet the specific requirements of students. Indicators relating to the choice of land and the construction phase have been removed. Other indicators not reported here have been maintained similar to the standards for SBTool[PT]-STP for Office Buildings.

The SAHSB[PT] methodology was applied in Francisco de Holanda high school, Guimarães, Portugal, to verify the adequacy of the benchmarks and their practical applicability to the context of school buildings in Portugal. The results are good in most of the indicators and categories and the overall sustainability level (SL) is A.

## 2. Structure of the Methodology

The selection of indicators was made through a process in which a specialist chose which indicators and categories should compose the SAHSB[PT] methodology, as well as their weights, after considering all the indicators from LEED BD + C School, BREEAM Education, SBTool for K–12 Schools, and SBTool[PT]-STP for Office Buildings.

The SBTool[PT] methodology was chosen, because it allows a greater possibility for adjusting the indicators evaluated in each dimension, considering the building typology, building site, and its practicality, demonstrating greater flexibility and also reducing subjectivity, which makes it more adaptable in relation to other more rigid and extensive systems [1]. Furthermore, because the SBTool[PT] was developed in Portugal, it is the tool most suited to practical application in Portuguese society.

The evaluation process used in the SAHSB$^{PT}$ methodology follows the calculation methods of SBTool$^{PT}$-STP for Office Buildings, using indicators related to social, economic, and environmental aspects and comparing them with national practices [10].

Such indicators usually follow the following structure [1,11]:

1.  Objective aspects that are intended to promote and reward the building;
2.  Required elements for evaluation, in which the necessary data for the evaluation are found;
3.  The life cycle phase to which the indicators apply: project phase, construction phase, or use phase;
4.  Benchmarks of the indicators, corresponding to the levels of current practice and best practice;
5.  Method of calculation, which provides a description of the steps and methodologies needed to quantify the performance of the building;
6.  Standardization to normalize and quantify the performance level of the building in the context of the indicators according to the previously defined benchmarks;
7.  Evaluation to summarize the performance level of the building at the indicator level, using the qualitative scale consisting of six levels (A+–E), which is used to communicate the result of the evaluation.

The process of evaluating the sustainability of a high school building using the methodology proposed in this work, which is composed of three phases and mirrors the SBTool$^{PT}$ evaluation framework [1,11].

**Phase 1:**

The evaluation of the specific performance of each indicator includes the following steps: quantification of the indicators and their standardization.

The standardization of the indicators sets a dimensionless value that demonstrates the performance of the building in relation to the benchmarks. In this process, the Diaz–Baltero equation is used [12].

$$\overline{P_I} = \frac{P_I - P_{I*}}{P_{I*} - P_{I*}} \tag{1}$$

where $\overline{P_I}$ represents the standardization value of the indicator; $P_I$ represents the result of the quantification of the indicator; $P_{I*}$ represents the value of the best practice; and $P_{I*}$ represents the value of the conventional practice.

The best practice is the value corresponding to the data originated for scientific work concerning the subject of each indicator, data produced by designers who already have some reputation in the field of sustainable construction, or data developed as a result of existing policies and standards. In the absence of references for Portugal, data from other countries was used [1,11].

The standard practice is the minimum value acceptable to consider a building sustainable. This value is based upon the minimum levels prescribed in the regulations and requirements of the current construction or practices of construction of residential buildings in Portugal.

The standardization values (X) are converted into a qualitative scale to facilitate the understanding of the results obtained. These results range between A (more sustainable) and E (less sustainable). In the qualitative scale presented, level A corresponds to the best practice and D to the standard practice, as shown in Table 1 [1,11].

**Table 1.** General evaluation of dimension, categories, and indicators, [1,4] for the Sustainability Assessment of High School Buildings in Portugal (SAHSB$^{PT}$) methodology.

| Level | Conditions | Please Check the Level Reached (✓) |
|:---:|:---:|:---:|
| A$^+$ | X > 1.00 | |
| A (best practice) | 0.70 < X ≤ 1.00 | |
| B | 0.40 < $\overline{X}$ ≤ 0.70 | |
| C | 0.10 < X ≤ 0.40 | |
| D (standard Practice) | 0 ≤ X ≤ 0.10 | |
| E | X < 0 | |



**Phase 2:** Quantification of the specific performance in terms of categories, dimensions and sustainability level.

Categories—indicators are aggregated into 11 different categories, summarizing construction performance. These categories are "Climate change outdoor air quality" (C1); "Biodiversity/land use" (C2); "Energy" (C3); "Materials, solid residues and resources, management" (C4); "Water" (C5); "User health and comfort" (C6); "Accessibility" (C7); "Security and safety" (C8); "Education sustainability awareness" (C9); "Sustainability of area" (C10); and "Life cycle costs" (C11).

Dimensions—the categories are combined to summarize the performance of the building in three dimensions: environmental, social and economic dimensions.

Sustainability level—this is the overall performance of the building. The value of the global performance of a building should not be used alone to report the sustainability of a building. It must be considered together with the results of the dimensions [1,4].

**Phase 3:** Completion of the Sustainability Certificate

At this stage, the certification of the building, for which the application was submitted, is demonstrated. This certificate reports the value of the global performance and the value of each dimension.

The most commonly used way to demonstrate the result of a building's sustainability assessment in a certificate is through a graduated scale that demonstrates the performance of the building and its level of performance in relation to the benchmarks [13]. The performance result should be demonstrated in a clear way, facilitating its interpretation and understanding by building users and being easily conceptualized by the evaluators.

The communication of sustainability through several indexes helps in the measurement, interpretation, and comparison of the building's performance, thus facilitating the identification and location of any problems and, with that, their solutions. Thus, the categorization of performance levels is achieved by using a six-level scale: from A + (more sustainable) to E (less sustainable), in which A corresponds to the best practice and D to conventional practice according to the SBTool[PT] H methodology.

The sustainability certificate shows that the results obtained in the evaluation through the SAHSB[PT] methodology are composed of three fields:

1.  Identification of the school;
2.  Sustainability level—sustainability level of the evaluated building and its performance in the three dimensions of sustainable development;
3.  Disaggregation of performance by each category—performance of the building at the level of each of the 11 categories considered in the methodology.

## 3. Materials and Methods

Several indicators of SBTool[PT]-STP for Office Buildings were taken or adapted from SBTool[PT], which, in turn, were based on the international SBTool, LEED, and BREEAM methodologies. In the evaluation of school buildings, some similar indicators were also used, and some indicators of SBTool[PT]-STP for Office Buildings were maintained in SAHSB[PT], because office buildings and school buildings share similar characteristics. Several aspects justify the importance and permanence of these indicators in the SAHSBPT methodology, as follows:

Indicator 1—Life cycle environmental impacts: this indicator aids the construction to improve the quality of the outdoor air and to prevent climate change, which reduces the environmental impact;

Indicator 2—Heat island effect: this indicator supports the use of materials in roofs and façades that decrease the heat island effect;

Indicator 3—Land use efficiency: this indicator supports the mitigation of the size of land used for construction, intending to preserve the environment;

Indicator 4—Certificated materials: this indicator supports the use of wood or organic products in building constructions, so that they have environmental certifications, thus decreasing deforestation;

Indicator 5—Energy consumption: this indicator supports the decrease of energy consumption in buildings, through the use of efficient equipment and passive solutions;

Indicator 6—Renewable Energy: this indicator supports the use of energy from renewable sources, decreasing the consumption of energy produced from fossil sources;

Indicator 7—Commissioning: this indicator supports the proper management of all the mechanical systems of a building over its life cycle, promoting the mitigation of environmental impacts, costs and energy consumption, optimizing the functionality of the building;

Indicator 8—Reuse and Recycle of Materials: this indicator supports the use of reused and recycled materials in new constructions, decreasing the cost and the energy necessary to produce new materials;

Indicator 9—Environmental management plan: this indicator assists in the management and control of the entire property, detecting the malfunction of several sectors, solving problems in a faster and efficient way, avoiding financial and environmental damages;

Indicator 10—Flexibility and adaptability: this indicator supports the use of construction solutions and constructive processes that facilitate changes in the purposes of the building, in repair works or in dismantling. Therefore, it decreases the need to use new construction materials, reducing costs and environmental impacts;

Indicator 11—Water consumption: this indicator supports the decrease in water consumption inside buildings and surrounding areas, therefore, it reduces costs and the environmental impacts related to water consumption;

Indicator 12—Water treatment and recycling: this indicator supports the reuse of water inside buildings and in the building's surroundings, using recycling devices, groundwater or gray water. Therefore, it reduces the cost and the environmental impacts related to water consumption;

Indicator 13—Collection and reuse of rainwater: this indicator supports the reuse of water inside buildings and in the land where the building is located, using aquifer recharge and collective tip flow in rainwater (RW) drainage systems. Therefore, it reduces the cost and the environmental impacts related to water consumption;

Indicator 14—Indoor air quality: this indicator supports the adequate level of indoor air quality to maintain health and the well-being of the users;

Indicator 15—Thermal comfort: this indicator aims to promote the thermal comfort of users inside school buildings.

Indicator 16—Visual comfort: this indicator supports the adequate level of visual comfort to maintain the health and well-being of the users;

Indicator 17—Acoustic comfort: this indicator supports the adequate level of acoustic comfort to maintain the health and well-being of the users;

Indicator 18—Ergonomic comfort: this indicator supports the adequate level of ergonomic comfort to maintain the health and well-being of the users;

Indicator 19—Mobility plan: this indicator supports the use of bicycles and the walking, and procedures that allow adequate mobility for people with some degree of difficulty in mobility, enlarging the quality of life of the students and employees of a school;

Indicator 20—Occupants security and safety: this indicator supports the implementation of measures to ensure the safety and security of the students in different aspects;

Indicator 21—Sustainability awareness: this indicator supports the increase of sustainability awareness among students, assisting them to follow sustainable practices in their routines and identify how to behave in a critical way with relation to sustainability issues;

Indicator 22—Accessibility to public transport: this indicator supports the use of public transportation by students, enlarging the quality of life of the students and employees of a school and reducing the pollution produced by private vehicles;

Indicator 23—Life cycle costs: this indicator supports the design and maintenance of school buildings with low life cycle costs;

The main features and categories in SBTool$^{PT}$ STP for Office Buildings were discussed. Several of these categories, some with adaptations, are used in the methodology developed in this work—the SAHSB$^{PT}$ methodology.

The indicators from the SBTool$^{PT}$-STP for Office Buildings that were kept in SAHSB$^{PT}$ are the following: "life cycle" (I1); "heat island effects" (I2); "land use efficiency" (I3); "certified materials" (I4); "commissioning" (I7); "environmental management plan" (I9); "flexibility and adaptability" (I10); "water consumption" (I11); "water treatment and recycling" (I12); "storm water management" (I13); "mobility plan" (I19); "accessibility to public transport" (I22); and "life cycle costs' (I23).

Other indicators were modified for several reasons, as explained below.

The indicators of "energy consumption" (I5) and "renewable energy" (I6) had to be modified due to a 2013 change in the Portuguese legislation regarding energy. The SBTool$^{PT}$-STP for Office Buildings was developed based on the Regulation for Energy Systems and Air Conditioning in Buildings (RSECE) [14] and on the Regulation of the Thermal Performance of Buildings (RCCTE) [15]. These regulations were replaced in 2013 by the Regulation of the Energy Performance of Residential Buildings (REH) [16] and the Energy Performance Regulation of Trade and Service Buildings (RECS) [17].

The indicator of "reuse and recycle of materials" (I8) is based on the indicators "reused materials" (I10) and "recyclable materials" (I11) in SBTool$^{PT}$-STP for Office Buildings. These indicators exist to promote more frequent consideration related to this subject. In addition, the choice of recycled or recyclable building materials is made in the same phase of the project, which explains the union of these two indicators.

Regarding indicators concerning the comfort of the user, such as "indoor air quality" (I14); "thermal comfort" (I15); "visual comfort" (I16); and "acoustic comfort" (I17); the SBTool$^{PT}$-STP for Office Buildings was used as the basis. However, the scope of the indicators was modified, because SAHSB$^{PT}$ aims to serve the needs of children who remain confined for long periods in a learning environment; thus, some standards of comfort must be stricter.

In this century, changes to high school buildings are usually made through reforms, with very few new constructions. The indicators of "sustainable location" (I4); "local biodiversity protection during construction"; and "waste management indicator" (I12), which belong to the SBTool$^{PT}$-STP for Office Buildings, relate to the construction phase, and consequently, they were not included in the SAHSB$^{PT}$ methodology.

Additionally, a new indicator was also included: "ergonomic comfort" (I18). Through studies about ergonomic indicators, such as that of Grandjean [18], it has been observed that classroom desks of inappropriate sizes and shapes cause pain. Therefore, the need to include this indicator in the SAHSB$^{PT}$ methodology was determined.

The indicator of "occupants security" (I24) in SBTool$^{PT}$-STP for Office Buildings is denominated "occupants security and safety" (I20) in the SAHSB$^{PT}$ methodology. The issue concerning occupant safety in the school environment can be evaluated according to different aspects. In the SBTool$^{PT}$-STP for Office Buildings, this question is determined by aspects of the guaranteed operating conditions of the main building services, such as water, energy, and telecommunications, as well as the safety of users, including proper lighting and surveillance systems. These factors constitute part of the methodology that still needs to be developed.

Issues concerning accident prevention have also been addressed, and factors such as the presence of materials that can be dangerous in the high school environment, safety in the area of sports and recreation, and safety on the stairs, among others, were analyzed. Through studies about safety in high school buildings, including those of Jacob [19] and Zwetsloot [20], it was realized that major accidents occur in the area of sports and recreation or on the stairs; therefore, the need to include this indicator in the SAHSB$^{PT}$ methodology was determined.

The indicator of "sustainability awareness" (I21) was added to the methodology to be developed. This indicator seeks to inform school students about the importance of sustainability in today's world. Increasing students' awareness of the importance of sustainability helps to reduce the environmental impact produced by school buildings.

*Categories and Indicators*

The main objective of these indicators is to quantify, simplify, and communicate some characteristics of the buildings. The combination of several indicators forms a category. There are large differences between the indicators used in different methodologies for evaluating the sustainability of school buildings due to the economic, environmental, and socio-cultural differences and the technologies in each country [3]. The LEED (US), BREEAM (UK), and SBTool (Canada) methodologies were initially developed according to the realities of their original countries, and later, they were adapted to the circumstances of each country in which they were applied.

In this section, all the indicators and categories of the SAHSB$^{PT}$ are presented together with an explanation of why they were chosen and a comparison with similar indicators belonging to other methodologies applicable to schools, such as LEED BD + C Schools, SBTool International for School, and BREEAM Education.

### C1. CLIMATE CHANGE AND OUTDOOR AIR QUALITY

Over the years, excessive $CO_2$ emissions have given rise to climate change. It is necessary to seek new proposals to reduce $CO_2$ emissions, in order to avoid or reduce the consequences of this phenomenon. This category includes two indicators: "life cycle environmental impacts" (I1) and "heat island effect" (I2).

#### Indicator 1—Life cycle environmental impacts

The objective of this indicator is to encourage and promote the use of materials and construction solutions that have low environmental impact.

It was not possible to investigate the constructive solutions used in school buildings in Portugal, because it was not viable to carry out such an exhaustive study. The calculation method used for this indicator is the same as that used in the SBTool$^{PT}$-STP for Office Buildings and SBTool$^{PT}$-H [3]; therefore, this indicator is similar to those used in these previous methodologies.

The same database, reference practice, and method of calculation were used in this research in order to maintain the standard of SBTool methodologies adapted to Portugal. According to these methodologies, first, the environmental impacts of the life cycle of the building are quantified, and then, these impacts are quantified with respect to the reference building.

#### Indicator 2—Heat island effect

The objective of this indicator is to promote the use of green areas located in the outer area of the building and materials with high reflectance to reduce the heat island effect in urban areas. Portugal maintains schools built in different historical periods; nevertheless, using a significant variety of materials and constructive processes, their façades are usually made to be white or other light tones, and their roofs are usually made with ceramic or metallic tiles. These materials support the reduction of heat island effect [21].

Currently, the construction industry favors the use of some parameters to predict the thermal behavior of materials used in façades when subjected to solar radiation, considering the interaction between different properties of the materials without using complex calculations, such as the Solar Reflectance Index (SRI) [21]. The SRI is used in the SAHSB$^{PT}$ methodology, for its effective representation of the thermal behavior of a constructed surface subjected to solar radiation and for its ease of calculation. This indicator is similar to those used in the SBTool$^{PT}$-STP for Office Buildings and SBTool$^{PT}$-H.

### C2. BIODIVERSITY AND LAND USE

The growth of areas used for agriculture, the destruction of forests, and the increase of urban areas are exacerbating negative environmental impacts. To minimize them, buildings must be constructed in areas that have already been occupied by other buildings. This category includes two indicators: "land use efficiency" (I3) and "certificated materials" (I4).

#### Indicator 3—Land use efficiency

The objective of this indicator is to support the decrease of the impact caused by the growth of urban areas by optimizing land occupation, where the construction of buildings is allowed, making the best use of these constructed areas. The references practices and the method of calculation for

this indicator are the same as those used in SBTool$^{PT}$-STP for Office Buildings. Thus, this indicator is similar to those used in the SBTool$^{PT}$-STP for Office Buildings and SBTool$^{PT}$-H.

This indicator in the SAHSB$^{PT}$ methodology considers the total area of the school complex, the area resulting from the vertical projection on the ground of buildings, the land area in vertical projection, the sum of the school buildings' compartment areas, and the number of classroom occupants to calculate the normalized value of the Index of Territorial Efficiency Ratio Occupation of the building.

### Indicator 4—Certificated materials

The objective of this indicator is to promote the use of organic products with environmental certification. There are many Portuguese and international laws regarding certified materials. The reference practices and the method of calculation for this indicator are the same as those used in the SBTool$^{PT}$ for Office Buildings. Thus, this indicator is similar to that used in the SBTool$^{PT}$-STP for Office Buildings. This indicator is important for schools, because, in addition to having windows, doors, and furniture, which are common in other types of buildings, there are many school desks, which are made of wood.

This indicator in the SAHSB$^{PT}$ methodology is concerned with the cost of wood or organic materials with environmental certification, such as stairs, doors, windows, coatings for floors, walls, ceilings, fixed furniture, footers, and structural elements made of wood.

### C3. ENERGY

Energy control, with the goal of reducing environmental impacts, is important in the proposed methodology. Energy consumption occurs during all phases of the life cycle of a building, starting with the manufacture of materials and continuing during the use phase. This category includes three indicators: "energy consumption" (I5); "renewable energy" (I6); and "commissioning" (I7).

### Indicator 5—Energy consumption

The objective of this indicator is to support the decrease of energy consumption in educational buildings through the use of passive solutions and efficient equipment. This indicator in the SAHSB$^{PT}$ methodology is based on the RECS [17] procedures for energy consumption and is concerned with values of the energy consumption per year related to electricity (EE) and gas (EG) (Energy Certificate or the Thermal Engineering Project). Standard practice and best practice are based on the calculation sheet delivered with the Regulatory Entity and the Energy Certificate. This indicator is different from any used in SBTool$^{PT}$-STP for Office Buildings, because it is based on RSECE [15] and RCCTE [16].

### Indicator 6—Renewable Energy

Similar to Indicator 6, the objective of this indicator in the SAHSB$^{PT}$ methodology is to support the decrease of energy consumption in buildings using passive solutions and efficient equipment. The assessment is based on the procedures established by RECS for the renewable energy of the building. This indicator is different from any used in the SBTool$^{PT}$-STP for Office Buildings, because it is also based on RSECE [8] and RCCTE [9].

### Indicator 7—Commissioning

The objective of this indicator is to promote the appropriate management of mechanical systems throughout the building life cycle. The best practice and the method of calculation for this indicator are the same as those used in SBTool$^{PT}$-STP for Office Buildings. Thus, this indicator is similar to that used in SBTool$^{PT}$-STP for Office Buildings. This indicator in the SAHSB$^{PT}$ methodology is assessed using a form that evaluates the commission team; the documentation regarding energy and the building system; the schedule that defines the important dates and milestones and the budget for the purchase and installation of energy generation for the building; the plan for the management of mechanical systems; and performance verification.

### C4. MATERIALS, SOLID RESIDUES, AND RESOURCES MANAGEMENT

The construction industry is one of the largest generators of solid waste. The production of waste occurs during construction, demolition, and use by the occupants. This category includes three indicators: "reuse and recycle materials" (I8); "environmental management plan" (I9); and "flexibility and adaptability" (I10).

**Indicator 8—Reuse and Recycling of materials**

The objective of this indicator is to support and reward the use of reused materials and the use of materials with recycled content in the building. This indicator is the intersection of the indicators of "reused materials" (I10) and "recyclable materials" (I11) used in the SBTool$^{PT}$-STP for Office Buildings. This indicator in the SAHSB$^{PT}$ methodology concerns the value corresponding to the sum of the cost of the materials or elements in the construction that are pre-existing in the building and will be reused and the materials or construction elements from deconstructions located outside of the site that will be reused. The reference practices and method of calculation for this indicator are the same as those used in SBTool$^{PT}$-STP for Office Buildings.

**Indicator 9—Environmental management plan**

The objective of this indicator is to support the appropriate management of resources during the use phase of the building and/or the adoption of an Environmental Management System (EMS).

This indicator in the SAHSB$^{PT}$ methodology is assessed using a form that evaluates the environmental management system; monitoring systems; product consumption in the use phase; and the training of occupants. This indicator is similar to that used in the SBTool$^{PT}$-STP for Office Buildings.

**Indicator 10—Flexibility and adaptability**

The objective of this indicator is to promote the adoption of construction solutions and processes that facilitate the changing uses of the building in repair work and decommissioning. Flexibility and adaptability are new subjects that have not been used widely in the planning of high school building construction.

This indicator in the SAHSB$^{PT}$ methodology is assessed using a form that evaluates the modularity of compartments, ventilation systems and air conditioning (duct location and size of equipment), water system and plumbing (location), and electrical and communications system (duct location). The reference practices and method of calculation for this indicator are the same as those used in the SBTool$^{PT}$-STP for Office Buildings. This indicator is similar to those used in the SBTool$^{PT}$-STP for Office Buildings and SBTool$^{PT}$-H.

**C5. WATER**

The choice of suitable materials and equipment in the design phase helps to reduce water consumption. Water consumption should be reduced in the construction phase but especially in the use phase of a building. This category includes three indicators: "water consumption" (I11); "water treatment and recycling" (I12); and "collection and reuse of rainwater" (I13).

**Indicator 11—Water consumption**

The objective of this indicator is to support the decrease of water consumption inside buildings in the use phase, using efficient systems. In the school environment, the main concerns relate to water saving, mainly in lavatories, toilets, and showers.

This indicator in the SAHSB$^{PT}$ methodology is assessed using a form that evaluates the annual drinking water consumption and potable water consumption for irrigation and the average daily water consumption of each interior and exterior device. This indicator is similar to that used in the SBTool$^{PT}$-STP for Office Buildings.

**Indicator 12—Water treatment and recycling**

The objective of this indicator is to promote the decrease of water consumption inside buildings in the use phase, using recycling devices and reusing rainwater, groundwater, or greywater. It has low reference values and may increase with the updating of the methodology.

This indicator in the SAHSB$^{PT}$ methodology is concerned with annual per capita (l/year) use of devices that drain into the recycling system. The reference practices and method of calculation for this indicator are the same as those used in the SBTool$^{PT}$-STP for Office Buildings; therefore, this indicator is similar to that used in the SBTool$^{PT}$-STP for Office Buildings.

**Indicator 13—Collection and reuse of Rainwater**

The objective of this indicator is to ensure groundwater recharge and reduce the peak flow in rainwater drainage systems.

This indicator in the SAHSB^PT methodology is concerned with the annual per capita (l/year) value of the building potential for Rainwater management. The reference practices and method of calculation for this indicator are the same as those used in the SBTool^PT-STP for Office Buildings; thus, this indicator is similar to those used in the SBTool^PT-STP for Office Buildings.

### C6. USER HEALTH AND COMFORT

The productivity, health, and quality of life of building occupants are related to internal environmental quality. It is important to maintain comfort indicators in a sustainability assessment of school environments, because students spend long periods of time in the school building, and discomfort interferes with students' concentration, learning, and health [22]. This category includes five indicators: "indoor air quality" (I14); "thermal comfort" (I15); "visual comfort" (I16); "acoustic comfort" (I17); and "ergonomic comfort" (I18).

### Indicator 14—Indoor air quality

The objective of this indicator is to promote an adequate level of air quality inside the buildings.

This indicator in the SAHSB^PT methodology is concerned with the renewal rate of the building air and the finishing materials with low volatile organic compounds (VOC) content. The method of calculation for this indicator is the same as that in the SBTool^PT-STP for Office Buildings; however, the reference practices are different.

### Indicator 15—Thermal Comfort

The objective of this indicator is to promote and reward the existence of a comfortable thermal environment inside the building.

This indicator in the SAHSB^PT methodology is concerned with the level of thermal comfort, especially during the summer and winter seasons. The reference practices for this indicator are different than those used in the SBTool^PT-STP for Office Buildings; nevertheless, the method of calculation is the same. Schools that have a control system for temperature do not need to be evaluated by this indicator.

### Indicator 16—Visual Comfort

The indicator in the SAHSB^PT methodology considers the illumination levels provided by natural lighting in all areas of the building that are occupied by users and the illumination levels provided by artificial lighting in all the areas of the building that are occupied by users. The method of calculation for this indicator is similar to that used in the SBTool^PT-STP for Office Buildings, although the reference practices are different.

### Indicator 17—Acoustic Comfort

The objective of this indicator is to promote the adoption of solutions that enable a high level of acoustic comfort for occupants. This indicator in the SAHSB^PT methodology is concerned with the level of acoustic comfort with airborne sounds between classrooms or other interior rooms and abroad, percussion sounds, and reverberation time in classroom. The reference practices used for this indicator are different from those used in the SBTool^PT-STP for Office Buildings; however, the method of calculation for this methodology and the SBTool^PT-STP for Office Buildings are similar.

### Indicator 18—Ergonomic Comfort

The objective of this indicator is to promote and reward the existence of ergonomic comfort inside the classroom.

This indicator in the SAHSB^PT methodology is concerned with students' comfort in school desks. Responses to questionnaires regarding ergonomic comfort of students of schools in Guimarães and Juiz de Fora (Brazil) demonstrate the necessity of including this indicator in the sustainability assessment tools for school buildings [22]; thus, this is a new indicator developed for the SAHSB^PT methodology.

It is necessary to understand several educational issues in order to determine the relationship of furniture with ergonomic, technological, and pedagogical criteria. The psychological and physical comfort of students directly interferes with learning achievements. The determination of anthropometric criteria for the height of the backrest, seat, and dimensions of chairs and tables is essential. Different sizes of chairs and tables should be used (small, medium, and large) to meet the basic requirements of students with different statures, supporting the implementation of various activities in the classroom [23].

### C7. ACCESSIBILITY

It is essential to prioritize access to public transport in order to reduce the need for car use and the consequent negative impacts of car use. This category includes the indicator: "mobility plan" (I19).

#### Indicator 19—Mobility plan

The objective of this indicator is to promote a sustainable mobility plan. The reference practices for this indicator are based on those used in the SBTool$^{PT}$-SPT for Office Buildings. This indicator is similar to that used in the SBTool$^{PT}$-STP for Office Buildings.

This indicator in the SAHSB$^{PT}$ methodology is assessed using a form that evaluates conditions for access to the building on foot or by bike, other sustainable transport, and access for disabled people.

### C8. SECURITY AND SAFETY

A school building must be designed or remodeled in order to preserve the highest safety for children. Safety involves several aspects related to the safety of the equipment and the provision for protection in case of fire. This category includes one indicator: "occupants' security and safety" (I20).

#### Indicator 20—Occupants' security and safety

The objective of this indicator is to promote the execution of procedures to ensure the safety and security of occupants.

This indicator in the SAHSB$^{PT}$ methodology is assessed using a form that evaluates the assurance of continued operation of the main building services (energy, water, and telecommunications) and the security of the building users. The part of the form that relates to the main building services is the same as that used in the SBTool$^{PT}$-SPT for Office Buildings; however, the part of the form that relates to concern about accident prevention was developed for this methodology. Because there are no studies referring to these data related to accident prevention and it is of the utmost importance, the best and the standard practices have very high values.

### C9. EDUCATION FOR SUSTAINABILITY AWARENESS

The objective of this indicator is to promote the level of sustainability awareness of students of the evaluated school.

The importance of sustainability awareness in schools is to encourage students to have positive attitudes towards sustainability and educate citizens who will be more conscious about sustainability. This category includes one indicator: "sustainability awareness" (I21).

#### Indicator 21—Sustainability awareness

The objective of this indicator is to promote the level of sustainability awareness of students of the evaluated school.

This indicator in the SAHSB$^{PT}$ methodology is assessed using a form that evaluates the level of the students' awareness related to sustainability. The form evaluates the level of environmental interest of the students, the frequency of environmental issues mentioned in class, and the frequency with which students do something to protect the environment in their daily lives. Other issues addressed are environmental practices in students' homes and how students believe environmental issues should be addressed in high schools. This indicator encourages awareness of sustainability among students and promotes positive attitudes towards sustainability in the students' daily lives [24]. This is a new indicator developed for the SAHSB$^{PT}$ methodology.

### C10. SUSTAINABILITY OF THE AREA

The use of public transport, such as train and bus, to school reduces the use of private cars and therefore reduces $CO_2$ emissions, causing less environmental impact. This category includes one indicator: "accessibility to public transport" (I22).

#### Indicator 22—Accessibility to public transport

The objective of this indicator is to promote and enhance buildings that meet most of the travel needs of the users through the public transport system.

This indicator in the SAHSB$^{PT}$ methodology considers the travel time to each public transport stop, the waiting time for each public transport line, and the frequency of each public transport line close to the building entrance. The reference practices and method of calculation used for this indicator

are the same as those used in the SBTool[PT]-STP for Office Buildings; hence, this indicator is similar to that used in the SBTool[PT]-STP for Office Buildings.

### C11. LIFE CYCLE COSTS

The cost of a building must take into account the value related to its entire life cycle. Generally, the costs of sustainable construction are lower than traditional buildings, but this is not always obvious to construction professionals. This category includes one indicator: "life cycle costs" (I23)

### Indicator 23—Life cycle costs

The objective of this indicator is to maximize the initial costs of the building and reduced life cycle costs.

This indicator in the SAHSB[PT] methodology is concerned with building performance in terms of initial cost and the operating costs (energy and water consumption). The reference practices and method of calculation for this indicator are the same as those used in the SBTool[PT]-STP for Office Buildings; therefore, this indicator is similar to that used in SBTool[PT]-STP for Office Buildings.

## 4. Results and Discussion

Methodologies for sustainability assessment use different indicators for the evaluation and certification of buildings, according to the environmental, social, cultural, technological, and economic characteristics of the countries in which they are applied. Because this research is based on the SBTool[PT]-STP for Office Buildings, the parallel between the indicators used by the SBTool[PT]-STP for Office Buildings and LEED BD + C Schools, SBTool for Schools, and BREEAM Education is evident. Office buildings and school buildings are both service buildings, and for this reason, they have several indicators in common.

The parallels between the indicators used by SAHSB[PT], SBTool[PT]-STP for Office Buildings, LEED BD + C for Schools, SBTool for K12 Schools, BREEAM Education, and the methodology proposed in this work—the SAHSB[PT] methodology—can be seen in Table 2.

**Table 2.** List of indicators used by Sustainability Assessment of High School Buildings in Portugal (SAHSB[PT]), SBTool[PT]-STP for Office Buildings, Leadership in Energy and Environmental Design (LEED) BD + C Schools, Sustainable Building Tool (SBTool) for K–12 Schools, and Building Research Establishment Environmental Assessment Method (BREEAM) Education.

| Dimension | Category | ID | SAHSB[PT] | SBTool[PT] | LEED | SBTool | BREEAM |
|---|---|---|---|---|---|---|---|
| Environmental | C1. Climate change outdoor air quality | I1 | Life cycle environmental impacts | X | X | X | X |
| | | I2 | Heat island effects | X | X | X | |
| | C2. Biodiversity/land use | I3 | Land use efficiency | X | | X | |
| | | I4 | Certificated materials | X | X | | X |
| | C3. Energy | I5 | Energy consumption | X | X | X | X |
| | | I6 | Renewable energy | X | X | | |
| | | I7 | Commissioning | X | X | X | X |
| | C4. Materials, solid residues and resources management | I8 | Reuse and recycle materials | X | X | X | X |
| | | I9 | Environmental management plan | X | X | | X |
| | | I10 | Flexibility and adaptability | X | X | X | |
| | C5. Water | I11 | Water consumption | X | X | X | X |
| | | I12 | Water treatment and recycling | X | X | X | X |
| | | I13 | Collection and reuse of Rainwater | X | X | X | X |
| Social | C6. User health and comfort | I14 | Indoor air quality | X | X | X | X |
| | | I15 | Thermal comfort | X | X | X | X |
| | | I16 | Visual comfort | X | X | X | X |
| | | I17 | Acoustic comfort | X | X | X | X |
| | | I18 | Ergonomic comfort | | | | |
| | C7. Accessibility | I19 | Mobility plan | X | X | X | X |
| | C8. Security/safety | I20 | Occupants security and safety | X | | X | X |
| | C9. Education/Sust. awareness | I21 | Sustainability awareness | | X | | X |
| | C10. Sust. of area | I22 | Accessibility public transport | X | | X | X |
| Economic | C11. Life cycle cost | I23 | Life cycle costs | X | | X | X |

After the study and analysis of the indicators and categories used in other sustainability assessment methodologies specific to school buildings, some conclusions were drawn regarding the indicators and categories that would be better suited to the methodology proposed in this work, as demonstrated in Table 2.

The SAHSB$^{PT}$ evaluation guide reduces errors in the evaluation process, allowing the evaluator to quantify the performance of the building at the level of each indicator, category, or dimension, resulting in the overall performance of the building (sustainability level—NS).

## 5. Conclusions

The development of a methodology to assess the sustainability of school buildings that is specific to each country or region is necessary, because the high school environment is very particular. In addition, sustainability in a school building improves the performance, safety, and health of teachers, students, and staff. Increased attention to the construction, design, and operational practices of schools contributes to the achievement of national sustainability goals relating to the protection of the environment.

The SAHSB$^{PT}$ methodology proposed in this work supports the development of high school buildings that offer comfort to the users, as well as low environmental impacts at a moderate cost. The importance of this work lies in the adaptation of the SBTool methodology for sustainability assessment specific to high school buildings located in Portugal.

A good sustainability assessment tool, in addition to addressing environmental and economic concerns, influences the well-being of the building users, thus reducing negative impacts on their health and improving their quality of life. Moreover, there is a tendency of local legislation to follow the requirements of the methodology applied to assess sustainability in school buildings, especially when it is widely used by the population, thus bringing additional benefits to society.

**Author Contributions:** Conceptualization, T.S.S. and M.d.A.; Methodology, T.S.S.; Formal Analysis, M.d.A. and T.S.S.; Investigation, T.S.S.; Original Draft Preparation, T.S.S.; Review and Editing, M.d.A., T.S.S., and L.B.; Project Administration, T.S.S.; and Funding Acquisition, T.S.S.

**Funding:** This research did not receive any specific grant from funding agencies in the public, commercial, or non-profit sectors.

**Acknowledgments:** T.S.S. would like to thank the following: the administrative director of Francisco de Holanda High School, Abílio Ferreira, for allowing access to the school and always giving all the necessary information with great attention; and Parque Escolar Company (EPE) for supplying the necessary materials for the execution of this study.

**Conflicts of Interest:** The authors declare no conflict of interest.

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
