# Peer review of "Adaptation of the SBTool for Sustainability Assessment of High School Buildings in Portugal—SAHSBPT"

_applsci, doi:10.3390/app9132664_

Round 1

Reviewer 1 Report

The authors fulfilled all the requirements. 

Round 1

Reviewers’ Comments and Authors Responses

Reference: Adaptation of the SBTool for Sustainability Assessment of High School Buildings in Portugal—SAHSBPT - applsci-533668

Response to technical check results comments

The authors would like to thank the area editor and the reviewers for their precious time and invaluable comments. We have carefully addressed all the comments. The corresponding changes and refinements made in the revised paper are summarized in our response below.

Reviewer #1: The authors fulfilled all the requirements.

Reviewer #2:

Comment 1 - Better organize Table 2, avoiding typos (for instance, “Dimen Sion”, C2.Biodiversity”, etc.). At the same time, it is not clear why there some dashes in some cells of the Table and some cells are empty. Please check!

Response:  We accept this reviewer´s comment and we organized Table 2.

Reviewer 2 Report

The sustainability assessment of building project represents a significant. The paper is very well organized, and the results are clearly presented and discussed.

Only the following revision is recommended:

-       Better organize Table 2, avoiding typos (for instance, “Dimen Sion”, C2.Biodiversity”, etc.). At the same time, it is not clear why there some dashes in some cells of the Table and some cells are empty. Please check!

Round 1

Reviewers’ Comments and Authors Responses

Reference: Adaptation of the SBTool for Sustainability Assessment of High School Buildings in Portugal—SAHSBPT - applsci-533668

Response to technical check results comments

The authors would like to thank the area editor and the reviewers for their precious time and invaluable comments. We have carefully addressed all the comments. The corresponding changes and refinements made in the revised paper are summarized in our response below.

Reviewer #1: The authors fulfilled all the requirements.

Reviewer #2:

Comment 1 - Better organize Table 2, avoiding typos (for instance, “Dimen Sion”, C2.Biodiversity”, etc.). At the same time, it is not clear why there some dashes in some cells of the Table and some cells are empty. Please check!

Response:  We accept this reviewer´s comment and we organized Table 2.

This manuscript is a resubmission of an earlier submission. The following is a list of the peer review reports and author responses from that submission.

Round 1

Reviewer 1 Report

The paper presents a research on the adaptation of Sustainable Building Tool (SBTool) to high school buildings in Portugal (SAHSBPT). This topic is relevant for the journal. Besides, the methodology applied is interesting. However, various points need revisions taking into account the following comments:

- Introduction. There are only 19 references in the whole text. Please expand the introduction section in relation with the built environment.

- There are any kind of validation of the proposed tool?

- Do you consider interesting the application to actual case studies?

- Check the paper again for any possible misprints.

Reviewer 2 Report

General comments:

This paper addresses a methodology that is proposed for using the sustainable building tool for school buildings in Portugal. Whereas the paper provides the necessary information on criteria for school buildings, it fails to provide information on why it is best suited for Portugal and not any other country. It also fails to justify why this research has been done. The authors should first compose their research question and defend it and then proceed with the specification of criteria.

Specific comments:

Line 39: Reference 1 is already mentioned. Please remove.

Line 42-43: In what way is it possible to improve the sustainability of building construction?

Lines 57-58: Is there a different tool according to different methodologies? if yes then they have to be mentioned. Also since when has the validation process started and when is it expected to be finished?

Line 114: Please distinguish if bold numbers in brackets refer to references or the bullets below. If the later is true, please explain.

Line 157-175: Phase 2 is not described in a way that it's understandable. Please provide example on what are the 11 categories mentioned and how its connected with table 1. The same applies to phase 3 and phase 4. The authors need to rewrite the whole passage since what they're trying to do is not comprehensible.

Line 204: The authors need to explain why the proposed indicators are used before mentioning them. This paragraph is not written as an introduction to section 3.

Line 548: The authors should combine table 2 and table 3 into a unified table in order to check the differences.